

# Heterogeneous OH Oxidation of Isoprene Epoxydiol-Derived Organosulfates: Kinetics, Chemistry and Formation of Inorganic Sulfate

Hoi Ki Lam[1], Kai Chung Kwong[1], Hon Yin Poon[1], James F. Davies[2], Zhenfa Zhang[3], Avram Gold[3], Jason D. Surratt[3], Man Nin Chan[1, 4]*

[1]Earth System Science Programme, Faculty of Science, The Chinese University of Hong Kong, Hong Kong, CHINA
[2]Department of Chemistry, UC Riverside, Riverside, USA
[3]Department of Environmental Sciences and Engineering, Gillings School of Global Public Health, The University of North Carolina at Chapel Hill, Chapel Hill, USA
[4]The Institute of Environment, Energy and Sustainability, The Chinese University of Hong Kong, Hong Kong, CHINA

*Correspondence to*: Man Nin Chan (mnchan@cuhk.edu.hk)

**Abstract**

Acid-catalyzed multiphase chemistry of epoxydiols formed from isoprene oxidation yields the most abundant organosulfates (i.e., methyltetrol sulfates) detected in atmospheric fine aerosols. This potentially determines the physicochemical properties of fine aerosols in isoprene-rich regions. However, chemical stability of these organosulfates remains unclear. As a result, we investigate the heterogeneous oxidation of aerosols consisting of potassium 3-methyltetrol sulfate ester ($C_5H_{11}SO_7K$) by gas-phase hydroxyl (OH) radicals through studying the oxidation kinetics and reaction products at a relative humidity (RH) of 70.8 %. Real-time molecular composition of the aerosols is obtained by using a Direct Analysis in Real Time (DART) ionization source coupled to a high-resolution mass spectrometer. Aerosol mass spectra reveal that 3-methyltetrol sulfate ester can be detected as its anionic form ($C_5H_{11}SO_7^-$) via direct ionization in the negative ionization mode. Kinetic measurements reveal that the effective heterogeneous OH rate constant is measured to be $4.74 \pm 0.2 \times 10^{-13}$ cm$^3$ molecule$^{-1}$ s$^{-1}$ with a chemical lifetime against OH oxidation of $16.2 \pm 0.3$ days. Comparison of this lifetime with those against other aerosol removal processes, such as dry and wet deposition, suggests that 3-methyltetrol sulfate ester is likely to be chemically stable over atmospheric timescales. Aerosol mass spectra only show an increase in the intensity of bisulfate ion ($HSO_4^-$) after oxidation, suggesting the absence of functionalization processes is likely attributable to the steric effect of substituted functional groups (e.g. methyl, alcohol and sulfate groups) on peroxy–peroxy radical reactions. Overall, potassium 3-methyltetrol sulfate ester likely decomposes to form volatile fragmentation products and aerosol-phase sulfate radial anion ($SO_4^{\bullet-}$). $SO_4^{\bullet-}$ subsequently undergoes intermolecular hydrogen abstraction to form $HSO_4^-$. These processes appear to explain the compositional evolution of 3-methyltetrol sulfate ester during heterogeneous OH oxidation.



## 1 Introduction

Isoprene (2-methyl-1,3-butadiene, $C_5H_8$), emitted from terrestrial vegetation to the atmosphere, is the largest source of non-methane volatile organic compounds. Apart from enhancing urban ozone levels via photochemical oxidation initiated by gas-phase hydroxyl (OH) radicals (Chameides et al., 1988), isoprene-derived oxidation products can also

significantly contribute to the formation of secondary organic aerosol (SOA) (Carlton et al., 2009). Gas-phase photochemical oxidation of isoprene by OH radicals can produce isoprene-derived hydroxyhydroperoxides (ISOPOOH) in yields greater than 70% under low-nitrogen oxide ($NO_X$) conditions (Paulot et al., 2009). Further reactions of ISOPOOH with OH radicals yield large quantities of isomeric isoprene epoxydiols (IEPOX), which partition into aqueous sulfate aerosols through acid-catalyzed ring-opening reactions. This multiphase chemical

pathway is a key for the substantial production of isoprene-derived SOA constituents (e.g. 2-methyltetrols, $C_5$-alkene triols, organosulfates, 3-methyltetrahydrofuran-3,4-diols and oligomers) within atmospheric fine particulate matter ($PM_{2.5}$) (Carlton et al., 2009; Froyd et al., 2010; Surratt et al., 2010; Lin et al., 2012).

Among these SOA constituents, IEPOX-derived organosulfates (e.g. methyltetrol sulfates) have been widely

detected in atmospheric aerosols and are estimated to account for 0.3–1.7 % of the total organic carbon (Chan et al., 2010; Froyd et al., 2010; Hatch et al., 2011; Lin et al., 2012; Stone et al., 2012; He et al., 2014; Budisulistiorini et al., 2015; Rattanavaraha et al., 2016; Meade et al., 2016; Hettiyadura et al. 2017). While the formation mechanisms of organosulfates have been extensively studied (Surratt et al., 2007, 2008; Minerath et al., 2009; Cole-Filipiak et al., 2010; Nozière et al., 2010; Lin et al., 2012; Nguyen et al., 2014), their chemical transformations and stability

remain unclear. These low-volatility organosulfates are preferentially present in aerosol phase and can be oxidized by gas-phase oxidants (e.g., OH radicals, ozone, and nitrate radicals) at or near the aerosol surface throughout their atmospheric lifetimes. The heterogeneous oxidative processes can change the size, composition and physicochemical properties (e.g., light scattering and absorption, water uptake and cloud condensation nuclei activity) of both laboratory-generated and atmospheric organic aerosols (Rudich et al., 2007; George and Abbatt,

2010; Kroll et al., 2015). However, the extent of heterogeneous oxidation of organosulfates has not been clearly examined to date. Therefore, a better understanding of aerosol-phase transformations of isoprene-derived organosulfates can provide more insights on their potential impacts on human health, air quality and climate.

In this work, we investigate the heterogeneous OH oxidation of potassium 3-methyltetrol sulfate ester ($C_5H_{11}SO_7K$,

**Table 1**), as a single-component aerosol system by using an aerosol flow tube reactor at 70.8 % RH in order to gain a more fundamental understanding of the kinetics and chemistry. The molecular composition of the aerosols before and after oxidation is characterized in real-time using a soft atmospheric pressure ionization source (Direct Analysis in Real Time, DART) coupled to a high-resolution mass spectrometer. The 3-methyltetrol sulfate ester investigated in this study is one of the isomers of the methyltetrol sulfates found in atmospheric aerosols, which are collectively

the most abundant particulate organosulfates (Budisulistiorini et al., 2015). On the basis of aerosol mass spectra and previously reported reaction pathways, oxidative kinetics and reaction products resulting from the heterogeneous OH oxidation of 3-methyltetrol sulfate ester are discussed. We acknowledge that although 3-methyltetrol sulfate



ester derived from the reactive uptake of gas-phase δ-IEPOX onto sulfate seed aerosols is not the sole contributor to IEPOX-derived organosulfates (Cui et al., 2018), the findings of this work provide a basis for understanding better the heterogeneous OH reactivity of other IEPOX-derived organosulfates (e.g. 2-methyltetrol sulfate esters) that predominate in atmospheric aerosols.

## 2 Experimental Methods

The heterogeneous OH oxidation experiments were carried out in an aerosol flow tube reactor at 70.8 % RH. The synthesis of potassium 3-methyltetrol sulfate ester has been described in the literature (Bondy et al., 2018). The experimental details of the oxidation experiment have been explained elsewhere (Chim et al., 2017a, 2017b, 2018).

Briefly, 3-methyltetrol sulfate ester aerosols were generated by an atomizer (TSI, model 3076) and mixed with nitrogen, oxygen, ozone and hexane before entering the reactor. Inside the reactor, the aerosols were oxidized heterogeneously by gas-phase OH radicals, which were generated by the photolysis of ozone under ultraviolet light at 254 nm in the presence of water vapor. The RH within the reactor was controlled by varying the dry/wet gas ratio. A water jacket was used to maintain a stable temperature of 20 $^{\circ}$C inside the reactor. The OH concentration was

controlled by varying the ozone concentration. By measuring the decay of hexane using a gas chromatography coupled with a flame ionization detector, the OH exposure, which is an integral of gas-phase OH radical concentration and reaction time, can be calculated (Smith et al., 2009):

$$\text{OH exposure } = \int_o^t [\text{OH}]\text{dt} = -\frac{\ln\left(\frac{[\text{Hex}]}{[\text{Hex}]_0}\right)}{k_{\text{Hex}}} \qquad (1)$$

where $[\text{Hex}]_0$ and $[\text{Hex}]$ are the hexane concentration before and after OH oxidation, respectively, t is the reaction

time (or aerosol residence time), which was measured to be 1.3 min, $k_{\text{Hex}}$ is the rate constant for gas-phase OH reaction with hexane, and [OH] is the time-averaged OH radical concentration. The OH exposure varied from 0 to ~ $1.4 \times 10^{12}$ molecules cm$^{-3}$ s. An annular Carulite catalyst denuder and an activated charcoal denuder were used to remove ozone and gas-phase species from the aerosol stream leaving the reactor, respectively. A fraction of the aerosol stream was sampled by a scanning mobility particle sizer (SMPS, TSI, CPC Model 3775, Classifier Model

3081) to measure the aerosol size and number distribution. Before oxidation, the surface-weighted diameter of the aerosols was 225.9 ± 1.4 nm. The remaining flow was directed into a stainless steel tube heater at 380–400℃, where the temperature and aerosol residence time in the heater were sufficient to completely vaporize the aerosols. The gas-phase species were then directed into the ionization region, an open narrow space between a DART ionization source (IonSense: DART SVP) and an atmospheric inlet of a high-resolution mass spectrometer (ThermoFisher, Q

Exactive Orbitrap) for real-time chemical characterization (Chan et al., 2013). The DART ionization source was operated in a negative ion mode, with helium as the ionizing gas (Cody et al., 2005). Metastable helium atoms generated were responsible for ionizing the gas-phase species in the ionization region. 3-methyltetrol sulfate ester can be ionized via direct ionization (Block et al., 2010; Hajslova et al., 2011). Most recently, Kwong et al. (2018) have detected the ionic form of two organosulfates (sodium salts of methyl sulfate (CH$_3$SO$_4$Na) and ethyl sulfate

(C$_2$H$_5$SO$_4$Na)) using the DART ionization source in negative ionization mode. Mass spectra were scanned over a range of *m/z* 70–700. Each mass spectrum was averaged over a 2–3 minute sampling time with a mass resolution of





about 140,000. The mass spectra were analyzed using Xcalibar software (Xcalibar Software, Inc., Herndon, VA, USA).

Aerosol phase state (e.g. solid or aqueous droplet) is known to play an important role in governing the

heterogeneous kinetics and chemistry of organic aerosols (McNeil et al., 2008; Renbaum and Smith, 2009; Chan et al., 2014; Slade and Knopf, 2014; Zhang et al., 2018). Hygroscopicity of potassium 3-methyltetrol sulfate ester aerosols has not been experimentally determined. Recently, Estillore et al. (2016) have measured the hygroscopicity of a diverse set of organosulfates, including potassium salts of glycolic acid sulfate, hydroxyacetone sulfate, 4-hydroxy-2,3-epoxybutane sulfate and 2-butenediol sulfate as well as sodium salts of methyl sulfate, ethyl sulfate,

propyl sulfate and benzyl sulfate. According to Estillore et al. (2016), these organosulfate aerosols did not show a distinct phase transition but absorbed or desorbed water reversibly when the RH increased or decreased, suggesting that these organosulfate aerosols were likely to be aqueous when RH was above 10%. Based on the literature results, we assume that potassium 3-methyltetrol sulfate ester exhibits hygroscopicity similar to the potassium salts of organosulfates reported by Estillore et al. (2016) (e.g. potassium 4-hydroxy-2,3-epoxybutane sulfate, $C_4H_7SO_6K$)

and remains aqueous prior to oxidation.

Volatilization of 3-methyltetrol sulfate ester and the impact of ozone and UV light on the aerosol composition were investigated in the absence of OH. The intensity of parent ions with aerosols removed from gas stream was less than 5% of that measured in presence of 3-methyltetrol sulfate ester aerosols, suggesting that the volatilization of 3-

methyltetrol sulfate ester is insignificant. No reaction product was observed in the presence of ozone without UV light or in the absence of ozone with the UV light, suggesting that 3-methyltetrol sulfate ester is not likely to be photolyzed or react with ozone under our experimental conditions.

## 3 Results and Discussions

### 3.1 Aerosol Mass Spectra

Figure 1 shows the data measured before and after oxidation at 70.8% RH. Before oxidation (Fig. 1a), a dominant ion peak at *m/z* 215 is observed, which corresponds to the ionic form of potassium 3-methyltetrol sulfate ester ($C_5H_{11}SO_7^-$) (Table 1). After oxidation (Fig. 1b), the parent ion remains the most dominant ion peak at the maximum OH exposure of ~$1.4 \times 10^{12}$ molecules cm$^{-3}$ s. There is no significant change in the ion intensity except for bisulfate

ion (HSO$_4^-$; *m/z* 97). Figure 2 shows the evolution of HSO$_4^-$ against OH exposure at 70.8% which indicates that the intensity of HSO$_4^-$ increases with the OH exposure. In the following sections, the kinetics and chemistry will be discussed based on the aerosol mass spectra and aerosol-phase reactions previously proposed in the literature.

### 3.2 Oxidation Kinetics

Oxidation kinetics can be quantified by analyzing the parent decay of 3-methyltetrol sulfate ester against the OH exposure. Figure 3 shows the normalized decay of 3-methyltetrol sulfate ester against OH exposure. At the maximum OH exposure (~ $1.4 \times 10^{12}$ molecules cm$^{-3}$ s), ~ 45% of 3-methyltetrol sulfate ester is oxidized. The



decay of the 3-methyltetrol sulfate ester can be fitted with an exponential function to obtain an effective second order heterogeneous OH rate constant ($k$) through Eq. (2) (Smith et al., 2009):

$$ln \frac{I}{I_0} = -k \cdot [OH] \, t \tag{2}$$

where $I$ is the ion signal at a given OH exposure, $I_o$ is the ion signal before oxidation, [OH] is the concentration of
gas-phase OH radicals, and $t$ is the reaction time. The $k$ is determined to be $4.74 \pm 0.2 \times 10^{-13}$ cm$^3$ molecule$^{-1}$ s$^{-1}$ (Table 1). Based on the fitted $k$ value, the chemical lifetime of 3-methyltetrol sulfate ester against heterogeneous OH oxidation ($\tau$) can be estimated by Eq. (3):

$$\tau = \frac{1}{k[OH]} \tag{3}$$

where [OH] is the 24-hour averaged OH radical concentration of $1.5 \times 10^6$ molecules cm$^{-3}$. The chemical lifetime
against oxidation is calculated to be $16.2 \pm 0.3$ days. The estimated timescales are longer than those of other important aerosol removal processes, such as dry and wet deposition (~ 7–10 days) (Seinfeld and Pandis, 2016). In addition to heterogeneous oxidation, organosulfates can undergo hydrolysis to form polyols and sulfuric acid with rates depending on their molecular structure and aerosol acidity (Darer et al., 2011; Hu et al., 2011). According to Darer et al. (2011), primary isoprene-derived organosulfates are stable against hydrolysis, even at low pH while
secondary and tertiary organosulfates are less thermodynamically stable than primary organosulfates. Since 3-methyltetrol sulfate ester is a primary organosulfate (Table 1), it is unlikely to hydrolyze. With reference to the literature results and our new experimental observations, 3-methyltetrol sulfate ester may possibly be considered chemically stable against heterogeneous OH oxidation and hydrolysis over atmospheric timescales.

**3.3 Reaction Mechanisms**

Schemes 1–5 show the reaction mechanisms proposed for the heterogeneous OH oxidation of 3-methyltetrol sulfate ester based on the aerosol mass spectra (Figs. 1 and 2) and well-known aerosol-phase reactions previously reported in the literature (George and Abbatt, 2010; Kroll et al., 2015). Potassium methyltetrol sulfate ester likely dissociates and exists in its ionic form in the droplets. In the first oxidation step, the OH radical abstracts a hydrogen atom to
form an alkyl radical which quickly reacts with oxygen to form a peroxy radical. No formation of alcohol or carbonyl functionalization product can be observed in aerosol mass spectra (Fig. 1), regardless of initial hydrogen abstraction site. The absence of functionalization products in the aerosol mass spectra may suggest that fragmentation processes are likely to have occurred. One possible explanation is that the multiple functional groups in the 3-methyltetrol sulfate ester (e.g. hydroxyl, methyl and sulfate groups, or a combination of these functional
groups, depending on the initial OH reaction site), which are always located at positions vicinal to a peroxyl group, may sterically hinder the association of two peroxy radicals into the cyclic tetroxide intermediate proposed in the Russell mechanism (Russell, 1957) and Bennett–Summers reactions (Bennett and Summers, 1974). As the intermediate is essential for the formation of alcohol and carbonyl functionalization products, the steric effect may in turn favor the formation of alkoxy radicals (Cheng et al., 2015). Alkoxy radicals, once formed, may tend to undergo
fragmentation due to the presence of vicinal hydroxyl groups, which lower the activation energy required for the decomposition of the alkoxy radicals (Wiegel et al., 2015; Jimenez et al., 2009). To gain more insights into this process, calculations are preformed using a structural–activity relationship (SAR) model developed for the



decomposition of alkoxy radicals (Peeters et al., 2004; Vereecken and Peeters, 2009). For example, when the β-carbons are methyl groups (isopropoxyl radical), the barrier height $E_b$ for the decomposition is calculated to be 15.6 kcal mol$^{-1}$. When one β-carbon bears a hydroxyl group (1-hydroxyisopropoxyl radical), $E_b$ is lowered to 8.1 kcal mol$^{-1}$, implying a faster decomposition rate. However, the effect of vicinal sulfate on the decomposition of an
alkoxy radical has not been estimated.

The fragmentation of alkoxy radicals can yield fragmentation products (without sulfate group), smaller organosulfates and sulfate radical anions (SO$_4$$^{\bullet-}$), as shown in Schemes 1–5. However, like the functionalization products, these products have not been detected (Fig. 1). Only the intensity of HSO$_4$$^-$ increases after oxidation (Fig.
2). On the basis of the proposed reaction pathways (Schemes 1–5), one possibility is that fragmentation products are likely to be volatile and partition back to the gas phase. The high volatility of these products may contribute to the absence of fragmentation products in the aerosol mass spectra. Fragmentation processes can also yield smaller organosulfates, which have low volatilities and remain in the aerosol phase. In our previous study, we have demonstrated that small organosulfates, such as sodium methyl sulfate (CH$_3$SO$_4$Na) and sodium ethyl sulfate
(C$_2$H$_5$SO$_4$Na), the alcohol and carbonyl functionalization products of sodium ethyl sulfate can be detected by the DART ionization source (Kwong et al., 2018). In other words, if these small organosulfates are formed in a significant amount during the OH oxidation of 3-methyltetrol sulfate ester, the DART ionization technique should be able to detect them, which is not the case with reference to the aerosol mass spectra (Fig. 1). We postulate that upon oxidation the reaction intermediates may tend to decompose and eventually yield SO$_4$$^{\bullet-}$. SO$_4$$^{\bullet-}$ is a strong oxidant in
aqueous phase and can react with a variety of organic compounds (e.g. alcohols, ethers, alkanes and aromatic compounds) (Neta et al., 1988; Clifton and Huie, 1989; Padmaja et al., 1993). For the OH reaction with 3-methyltetrol sulfate ester, SO$_4$$^{\bullet-}$ can abstract a hydrogen atom from a neighboring organic molecule (e.g. unreacted 3-methyltetrol sulfate ester) to form HSO$_4$$^-$ (R1) or react with aerosol-phase water to yield a HSO$_4$$^-$ and an OH radical (R2) (Tang et al., 1988) as illustrated below. It is noted that SO$_4$$^{\bullet-}$ or OH radical recycled from R2 can react
with 3-methyltetrol sulfate ester, contributing to the secondary chain reactions.

$$C_5H_{11}O_7S^- + SO_4^{\bullet-} \rightarrow C_5H_{10}O_7S^{\bullet-} + HSO_4^- \tag{R1}$$
$$SO_4^{\bullet-} + H_2O \rightleftharpoons OH^{\bullet} + HSO_4^- \tag{R2}$$

Since 3-methyltetrol sulfate ester is unlikely to hydrolyze (Darer et al., 2011), the formation of HSO$_4$$^-$ upon OH oxidation could be best explained by the formation and subsequent reactions of SO$_4$$^{\bullet-}$. For instance, the OH-initiated
oxidation by hydrogen abstraction at the terminal secondary carbon site (Scheme 1) can yield a peroxy radical. The formation of alkoxy radicals through the self-reactions of two peroxy radicals is expected to be favored due to the steric effect induced by the presence of methyl and hydroxyl groups near the peroxyl group. The resultant alkoxy radical likely undergoes β-scission, attributing to the hydroxyl group located near the alkoxy group could lower the energy barrier height (Jimenez et al., 2009). It is postulated that an alkyl radical of methyl sulfate ion could be
formed after multiple reaction steps proposed in the reaction scheme. Our recent work on the heterogeneous OH oxidation of sodium methyl sulfate has demonstrated that further reactions of this alkyl radical can lead to the formation of a formaldehyde and a SO$_4$$^{\bullet-}$, which subsequently abstract a hydrogen atom to form HSO$_4$$^-$ (Scheme 5)



(Kwong et al., 2018). At the same time, fragmentation products formed during β-scission (e.g. formic acid and acetic acid) are volatile and likely to partition back to the gas phase.

## 4 Conclusions and Atmospheric Implications

This work investigates the kinetics of oxidation and molecular transformations of potassium 3-methyltetrol sulfate ester resulting from the heterogeneous OH oxidation. Kinetic measurements reveal that the chemical lifetime of 3-methyltetrol sulfate ester against heterogeneous OH oxidation and hydrolysis are longer than those against other aerosol removal processes, such as dry and wet deposition. 3-methyltetrol sulfate ester is potentially chemically stable over its atmospheric timescale. Aerosol mass spectra reveal that only the intensity of $HSO_4^-$ increases after oxidation, suggesting the dominance of fragmentation processes over functionalization processes. During oxidation, alkoxy radials are likely to be formed following hydrogen abstraction of 3-methyltetrol sulfate ester by OH radicals because the reaction intermediates in the Russell and Bennett–Summers reactions are sterically unfavorable. The alkoxy radicals subsequently fragment into volatile products and $SO_4^{\bullet-}$. The volatile fragmentation products tend to partition into gas phase, while $SO_4^{\bullet-}$ undergoes intermolecular hydrogen abstraction to form $HSO_4^-$ in the aerosol phase. Smaller organosulfates have not been observed, possibly due to the rapid continuous decomposition of the reaction intermediates proposed in Schemes 1–5. The absence of smaller organosulfates suggests that the oxidation of 3-methyltetrol sulfate ester is not a source of smaller organosulfates detected in atmospheric aerosols. Further investigations into whether large organosulfates yield smaller organosulfates upon heterogeneous OH oxidation are desirable. Aerosol mass spectra have revealed that the OH oxidation of 3-methyltetrol sulfate ester can lead to the formation of inorganic sulfate (e.g. $HSO_4^-$), in accord with our report that the heterogeneous OH oxidation of sodium methyl sulfate and sodium ethyl sulfate can lead to the formation of $HSO_4^-$ (Kwong et al., 2018). Given the high atmospheric abundance of organosulfates in atmospheric aerosols, further study of the contribution and transformation of organosulfates to inorganic sulfate through chemical reactions (e.g. heterogeneous oxidation, aqueous-phase oxidation and hydrolysis) is desirable. Methyltetrol sulfates are the most abundant isoprene-derived organosulfates measured in atmospheric $PM_{2.5}$ samples collected from isoprene-rich regions influenced by anthropogenic emissions (Budisulistiorini et al., 2015; Hettiyadura et al. 2017). Additional studies are required to better understand the role of the molecular structure (i.e. position of the methyl and sulfate group) on the kinetics and chemistry of methyltetrol sulfates and other organosulfates upon heterogeneous OH oxidation; in particular the effect on the formation of smaller organosulfates, volatile fragmentation products and inorganic sulfate, since the 2-methyltetrol sulfate ester and its isomers rather than the 3-methyltetrol sulfate ester investigated in this study predominate in atmospheric aerosols (Cui et al., 2018).

## 5 Acknowledgements

H. K. Lam, K. C. Kwong, H. Y. Poon and M. N. Chan are supported by the CUHK direct grant (4053281) and Hong Kong Research Grants Council (HKRGC) Project ID: 2191111 (Ref 24300516) and 2130626 (Ref 14300118). Synthesis of potassium methyltetrol sulfate was supported by the National Science Foundation (NSF) under



Atmospheric and Geospace (AGS) Grant 1703535. We would like to thank Kevin Wilson for his insightful comments on the reaction mechanisms proposed for the OH reaction with 3-methyltetrol sulfate ester.

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


**Table 1. Chemical Structure, properties, effective heterogeneous OH rate constant and atmospheric lifetime against OH radical of potassium 3-methyltetrol sulfate ester**

| Compounds | Potassium 3-methyltetrol sulfate ester |
|---|---|
| Structural Formula | <br> HO$\diagup$$\diagdown$$\diagup$OSO$_3^-$ K$^+$ with OH groups |
| Molecular Formula | C$_5$H$_{11}$SO$_7$K |
| Molecular Weight (g mol$^{-1}$) | 254.30 |
| Relative Humidity, RH (%) | 70.8 |
| Effective Heterogeneous OH Rate Constant, $k$ ($\times 10^{-13}$ cm$^3$ molecule$^{-1}$ s$^{-1}$) | $4.74 \pm 0.2$ |
| Atmospheric Lifetime against OH Oxidation (Days)[a] | $16.2 \pm 0.3$ |

[a] using a 24-hour average OH concentration of $1.5 \times 10^6$ molecules cm$^{-3}$



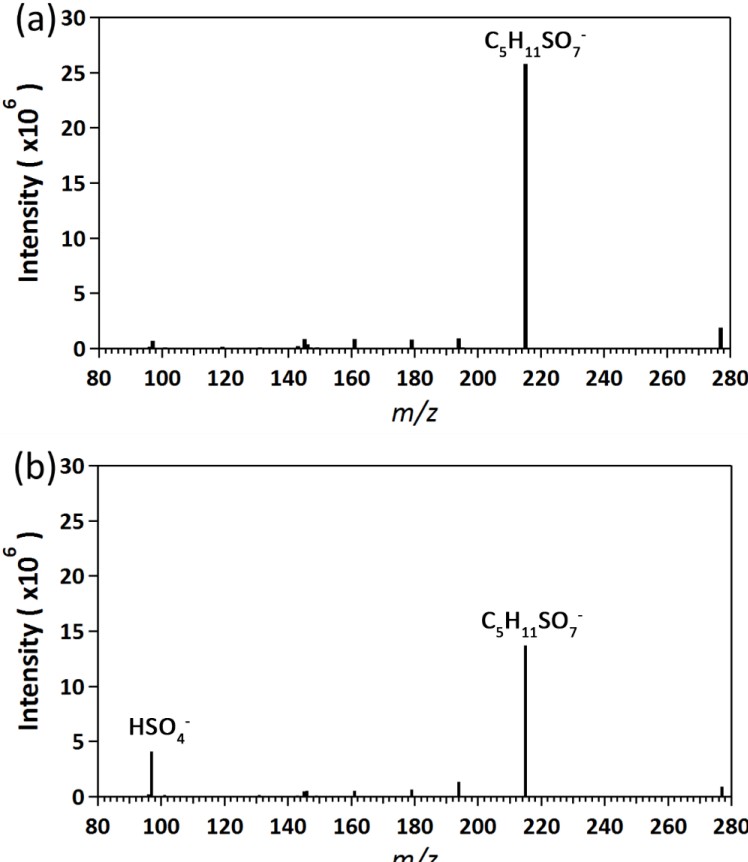

**Figure 1.** Aerosol mass spectra of potassium 3-methyltetrol sulfate ester before (a) and after (b) OH oxidation at 70.8% RH.



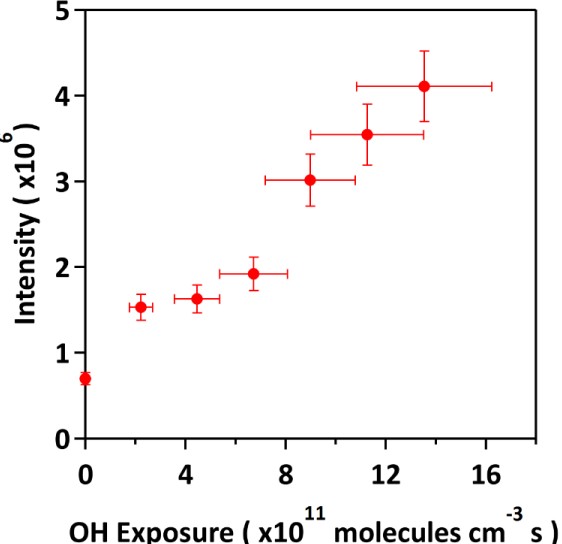

**Figure 2.** The evolution of the intensity of bisulfate ion ($HSO_4^-$) as a function of OH exposure during the heterogeneous OH oxidation of potassium 3-methyltetrol sulfate ester at 70.8% RH.

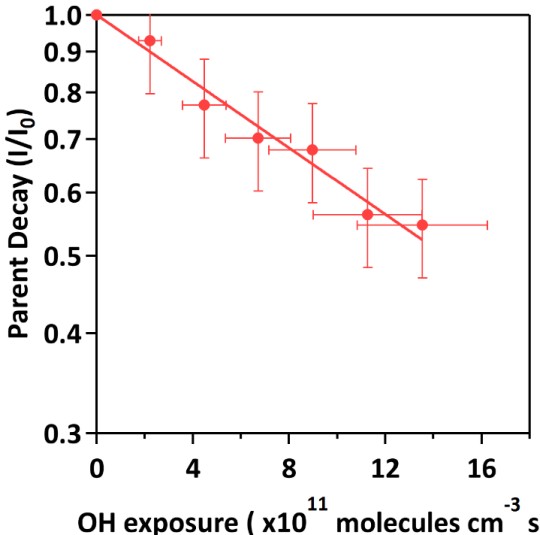

**Figure 3.** The normalized parent decay as a function of OH exposure during the heterogeneous OH oxidation of potassium 3-methyltetrol sulfate ester at 70.8% RH.





**Scheme 1.** Proposed reaction mechanisms for the heterogeneous OH oxidation of potassium 3-methyltetrol sulfate ester at the carbon labelled a.





**Scheme S5**

**Scheme 2.** Proposed reaction mechanisms for the heterogeneous OH oxidation of potassium 3-methyltetrol sulfate ester at the carbon labelled b.

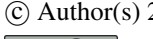



**Scheme 3.** Proposed reaction mechanisms for the heterogeneous OH oxidation of potassium 3-methyltetrol sulfate ester at the carbon labelled c.





**Scheme 4.** Proposed reaction mechanism for the heterogeneous OH oxidation of potassium 3-methyltetrol sulfate ester at the carbon labelled d.



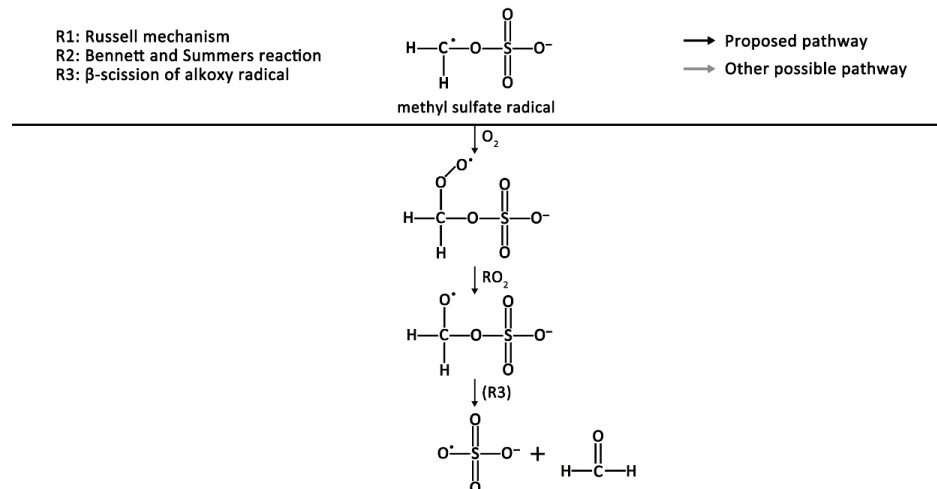

**Scheme 5.** Proposed reaction mechanism for the reactions of alkyl radical of methyl sulfate anion.