# Peer review of "Heterogeneous OH Oxidation of Isoprene Epoxydiol-Derived Organosulfates: Kinetics, Chemistry and Formation of Inorganic Sulfate"

_Atmospheric Chemistry and Physics, 2018_

## Referee Comment (RC1) · Anonymous Referee #3 · 17 Dec 2018

Review of "Heterogeneous OH Oxidation of Isoprene Epoxydiol-Derived Organosulfates: Kinetics, Chemistry and Formation of Inorganic Sulfate" (acp-2018-989).

This manuscript describes the OH oxidation of small organosulfates derived from atmospheric oxidation of isoprene. Using advanced mass spectrometry of aerosol particles generated in an aerosol flow reactor, it is shown that loss of 3-methyltetrol sulfate ester is slow, with a lifetime for oxidation of more than 2 weeks under likely atmospheric conditions. This suggests that other loss processes (e.g. deposition) may be the main atmospheric loss process for similar organosulfate esters. Lack of detection

of smaller oxidation products suggests formation of volatile products upon oxidation. These results are consistent with relatively high observed levels of organosulfates in atmospheric aerosol. This work is a solid contribution to the field, and I recommend this manuscript for publication pending minor revisions.

General Comments:

The experimental setup and data are fairly straightforward, generally giving clear, direct support of the key conclusion of a long atmospheric lifetime for small, oxidized organosulfates, but the detailed and long discussion of mechanistic details is not warranted. Without detection, in either gas or aerosol phase, of any reaction products other than bisulfate, the mechanistic discussion is largely speculative. Nearly all of this discussion should be removed. All reaction schemes presented should be moved to supplementary material and the mechanistic discussion must be reduced significantly. You cannot make such strong claims about a mechanism in which you more or less only observe loss of product with inability to detect products. Table 1 is the crucial result of this paper, not the speculation about the mechanism for oxidation.

You must comment on the potential for side-reactions, such as oxidation of second-generation products, due to the high OH levels used in this study.

Supplementary materials must include a figure showing time dependent variables during an experiment including: aerosol size distributions, hexane concentration, and product signal. This may be nearly a step function, but it should be clearly shown that your system clearly responds to initiation of oxidation.

Specific Comments

It should be noted that volatile products formed in the aerosol used here might remain in other aerosol if the Henry's law constant changes dramatically with composition, particularly if the volatile products are highly reactive with other types of aerosol species not present in your study (e.g. reactive nitrogen species). Comment on precedents

available in published literature.

Due to the open ionization region, is it possible that you lose volatile products as a result of detection method? This would mean that non-detection of products might be caused by the method of detection rather than evaporation under reasonable atmospheric conditions.

The non-detection of products (particularly from functionalization) using DART would be good information to add to the abstract.

Section 3.3

This should be reduced to a single paragraph that only mentions the proposed mechanism and does not go into details concerning issues such as the sterics of the reaction intermediates.

Page 2, Line 2 Change to read ", the largest atmospheric source"

4,6 Change to read "The hygroscpocity"

7,9 Change to read "stable over its atmospheric lifetime." 7, 12-13 Note that this is only a likely mechanism.

Figure 2 Y-axis should note m/z.

Figure 3 Y-axis should note m/z. Scheme 5. Only 1 pathway is shown, text seems to mention more than one.

---

## Referee Comment (RC2) · Anonymous Referee #1 · 23 Dec 2018

General Comments:

In this paper the authors describe results of a laboratory study of the heterogeneous oxidation of 3-methyltetrol sulfate ester by OH radicals in aqueous aerosol particles. This compound served as a surrogate for organosulfate esters formed from particle-phase reactions of isoprene oxidation products. Understanding the atmospheric fate of these compounds is important because of their potential importance as aerosol components. The reactions were conducted in a flow tube and the concentrations of the parent compound and products were monitored using DART mass spectrometry. The rates of

oxidation were quantified using relative rate methods and the measured rate constant used to estimate the lifetime of these and similar compounds in the atmosphere due to this reaction. The only major product observed was bisulfate, and a mechanism was proposed to explain its presence. The paper is clearly and concisely written and the experiments are straightforward. The interpretation is reasonable, although it is challenging to build a believable mechanism when only one of the proposed products was detected. I think the paper should be publishable after some comments are addressed, but there is clearly work to be done to provide more convincing evidence for the proposed mechanism.

Specific Comments:

Page 2, line 34: The kinetics method assumes that the particles are uniformly mixed, which is probably true here. But in the atmosphere where particles contain numerous compounds with a range of surface-active properties there will be significant inhomogeneities in surface concentrations where the OH reaction occurs, thus affecting estimated lifetimes. Some comments on this are warranted.

Page 5, line 20: The authors should justify their decision to ignore decomposition of a-hydroxyperoxy radicals to carbonyl + HO2, which is the dominant pathway in the gas phase. This should be based on kinetics arguments since the absence of product peaks in the mass spectrum could be influenced by ionization. Relevant rate constants are available [Neta et al., J. Phys. Chem. Ref. Data, 19, 413-513 (1990)].

Page 6, lines 13–18: I am not yet convinced by the argument that because the authors were able to detect small organosulfates in a previous study that they would have seen them here if they were present. How sensitive is the DART method to matrix effects, concentrations, and other conditions?

Page 7, lines 1–2 and elsewhere: Why wouldn't compounds that evaporate also be detected by DART, since the method for analyzing the particles involves first evaporating them at high temperature?

Technical Comments:

Page 3, line 15: Should be "chromatograph".

---

## Author Comment (AC1) · 21 Jan 2019

*This manuscript describes the OH oxidation of small organosulfates derived from atmospheric oxidation of isoprene. Using advanced mass spectrometry of aerosol particles generated in an aerosol flow reactor, it is shown that loss of 3-methyltetrol sulfate ester is slow, with a lifetime for oxidation of more than 2 weeks under likely atmospheric conditions. This suggests that other loss processes (e.g. deposition) may be the main atmospheric loss process for similar organosulfate esters. Lack of detection of smaller oxidation products suggests formation of volatile products upon oxidation. These results are consistent with relatively high observed levels of organosulfates in atmospheric aerosol. This work is a solid contribution to the field, and I recommend this manuscript for publication pending minor revisions.*

**We would like to sincerely thank the reviewer for his/her thoughtful comments and suggestions. Please see our responses to reviewer's comments and suggestions below.**

**Comment #1**

*The experimental setup and data are fairly straightforward, generally giving clear, direct support of the key conclusion of a long atmospheric lifetime for small, oxidized organosulfates, but the detailed and long discussion of mechanistic details is not warranted. Without detection, in either gas or aerosol phase, of any reaction products other than bisulfate, the mechanistic discussion is largely speculative. Nearly all of this discussion should be removed. All reaction schemes presented should be moved to supplementary material and the mechanistic discussion must be reduced significantly. You cannot make such strong claims about a mechanism in which you more or less only observe loss of product with inability to detect products. Table 1 is the crucial result of this paper, not the speculation about the mechanism for oxidation.*

**Author Response:**

We agree with the reviewer's comment. In the revised manuscript, we have reduced the mechanistic discussion and moved the detailed discussion and reaction schemes to the supplementary material. We also address that the reaction pathways (Schemes S1-S5) are tentatively proposed based on the observed aerosol-phase products detected by the DART. To better understand the reaction pathways, more studies are needed to characterize and quantify the reaction products in both aerosol and gas phases with different analytical techniques in order to cross check the aerosol composition data obtained by DART and verify the proposed reaction pathways in this work.

Page 5, Line 20,

" **3.3 Proposed Reaction Mechanisms**

Based on the aerosol mass spectra and well-known aerosol-phase reactions previously reported in the literature (George and Abbatt, 2010; Kroll et al., 2015), we tentatively propose reaction mechanisms for the heterogeneous OH oxidation of 3-methyltetrol sulfate ester. The reaction schemes proposed can be found in the supplementary materials (Schemes S1-S5). Briefly, potassium methyltetrol sulfate ester likely dissociates and exists in its ionic form in the droplets.

In the first oxidation step, the OH radical abstracts a hydrogen atom to form an alkyl radical which quickly reacts with oxygen to form a peroxy radical. We propose that the formation of alkoxy radical may be favored over the Russell mechanism (Russell, 1957) and Bennett–Summers reactions (Bennett and Summers, 1974) as functionalization products were not detected. Alkoxy radicals, once formed, may tend to undergo fragmentation due to the presence of vicinal hydroxyl groups, which lower the activation energy required for the decomposition of the alkoxy radicals (Cheng et al., 2015; Wiegel et al., 2015; Jimenez et al., 2009; Peeters et al., 2004; Vereecken and Peeters, 2009).

*Formation of HSO$_4^-$.* Sulfate radical anion (SO$_4^{\bullet-}$) can be formed through the decomposition of the alkoxy radical and is a strong oxidant in aqueous phase (Neta et al., 1988; Clifton and Huie, 1989; Padmaja et al., 1993). SO$_4^{\bullet-}$ can abstract a hydrogen atom from a neighboring organic molecule (e.g. unreacted 3-methyltetrol sulfate ester) to form HSO$_4^-$ (R1) or react with particle-phase water to yield a HSO$_4^-$ and an OH radical (R2) (Tang et al., 1988) as illustrated below. It is noted that SO$_4^{\bullet-}$ or OH radical recycled from R2 can react with 3-methyltetrol sulfate ester, contributing to the secondary chain reactions.

$$C_5H_{11}O_7S^- + SO_4^{\bullet-} \rightarrow C_5H_{10}O_7S^{\bullet-} + HSO_4^- \qquad\qquad (R1)$$

$$SO_4^{\bullet-} + H_2O \rightleftharpoons OH^\bullet + HSO_4^- \qquad\qquad (R2)$$

Since 3-methyltetrol sulfate ester is unlikely to hydrolyze (Darer et al., 2011), the formation of the HSO$_4^-$ upon OH oxidation could be best explained by the formation and subsequent reactions of SO$_4^{\bullet-}$.

Based on the proposed reaction mechanisms, the decomposition of alkoxy radicals can lead to formation fragmentation products (without sulfate group) and smaller organosulfates. We acknowledge that the ionization efficiency and detection limit of the reaction products are not fully understood. The absence of the potential products might attribute to the DART ionization and detection issues. More work is needed to investigate the formation and abundance of the reaction products formed in the aerosol and gas phases upon oxidation in order to better understand the reaction pathways. It is also noted that the formation of second or higher generation products are possible due to the high OH concentrations used in this study. While the potential secondary or higher generation products have not detected possibly due to their low concentrations and/or ionization issues, for clarify, the formation and oxidation of second or higher generation products are not further discussed."

**Comment #2**

*You must comment on the potential for side-reactions, such as oxidation of second-generation products, due to the high OH levels used in this study.*

**Author Response:**

We agree with the reviewer that first generation products can be further oxidized to form second or higher generation products due to the high OH levels in this study. However, the potential secondary or higher generation products have not detected possibly due to their low concentration and/or ionization efficiency. For clarity, the formation and oxidation of second or higher generation products are addressed but will not be discussed in details.

Page 6, Line 9, "It is also noted that the formation of second or higher generation products are possible due to the high OH concentrations used in this study. While the potential secondary or higher generation products have not detected possibly due to their low concentrations and/or ionization issues, for clarity, the formation and oxidation of second or higher generation products are not further discussed."

**Comment #3**

*Supplementary materials must include a figure showing time dependent variables during an experiment including: aerosol size distributions, hexane concentration, and product signal. This may be nearly a step function, but it should be clearly shown that your system clearly responds to initiation of oxidation.*

**Author Response:**

We would like to note that oxidation experiments were carried out in a flow tube reactor. We have measured the composition and size of the aerosols leaving the reactor before and after OH oxidation with a fixed reaction time (aerosol residence time is about 1.5 minutes in this study. Hence, the time dependent data (e.g. the time evolved data of aerosol size distributions and product signal) are not available in our study.

**Comments #4**

*It should be noted that volatile products formed in the aerosol used here might remain in other aerosol if the Henry's law constant changes dramatically with composition, particularly if the volatile products are highly reactive with other types of aerosol species not present in your study (e.g. reactive nitrogen species). Comment on precedents available in published literature.*

**Author Response:**

We agree with the reviewer that based on the proposed reaction mechanisms, volatile fragmentation products likely contain polar functional groups. They might partition back to the aerosols, for instance aqueous droplets because of their high water solubilities or Henry's law constants. Additionally, these fragmentation products could be reactively uptaken by the aerosols which contain reactive nitrogen or oxygen species through reactions. We have added this information in the revised manuscript.

Page 6, Line 29, "It is also noted that volatile fragmentation products likely contain polar functional groups. They may partition back to the aerosols, for instance aqueous droplets because of their high water solubilities or Henry's law constants. Additionally, they could be reactively uptaken by aerosols which contain reactive nitrogen or oxygen species through reactions."

**Comments #5**

*Due to the open ionization region, is it possible that you lose volatile products as a result of detection method? This would mean that non-detection of products might be caused by the method of detection rather than evaporation under reasonable atmospheric conditions.*

*The non-detection of products (particularly from functionalization) using DART would be good information to add to the abstract.*

**Author Response:**

As mentioned in the experimental section, the volatile fragmentation products or gas-phase species were removed by an activated charcoal denuder before entering the heater as well as ionization region, leaving only the particle-phase products (see Page 2 Line 22). We have clarified this point.

Page 3, Line 22, "An annular Carulite catalyst denuder and an activated charcoal denuder were used to remove ozone and gas-phase species from the aerosol stream leaving the reactor, respectively. As a result, only particle-phase products are detected. A fraction of the aerosol stream was sampled by a scanning mobility particle sizer (SMPS, TSI, CPC Model 3775, Classifier Model

3081) to measure the aerosol size and number distribution. The surface-weighted diameter of the aerosols was measured to be 225.9 ± 1.4 nm before oxidation. The remaining flow was then directed into a stainless steel tube heater at 380–400°C, where the temperature and aerosol residence time in the heater were sufficient to completely vaporize the aerosols."

**Comments #6**

*Section 3.3 This should be reduced to a single paragraph that only mentions the proposed mechanism and does not go into details concerning issues such as the sterics of the reaction intermediates.*

**Author Response:**

The reviewer has raised the same concern in **Comment #1.** We agree with the reviewer's comment. In the revised manuscript, we have reduced mechanistic discussion and moved the reaction schemes to the supplementary materials. We also address that the reaction pathways (Schemes S1-S5) are tentatively proposed based on the observed products detected by the DART. Please see our response to **Comment #1**. We have also removed the content about the steric effects on the formation of reaction products from the Abstract.

**Comments #7**

*Page 2, Line 2 Change to read ", the largest atmospheric source"*

**Author Response:**

We have revised the sentence in the manuscript.

Page 2, Line 2, "Isoprene (2-methyl-1,3-butadiene, $C_5H_8$), emitted from terrestrial vegetation to the atmosphere, is the largest atmospheric source of non-methane volatile organic compounds."

**Comments #8**

*Page 4, Line 6, Change to read "The hygroscpocity"*

**Author Response:**

We have revised the sentence in the manuscript.

Page 4, Line 5, "The hygroscopicity of potassium 3-methyltetrol sulfate ester aerosols has not been experimentally determined."

**Comments #9**

*Page 7, Line 9, Change to read "stable over its atmospheric lifetime."*

**Author Response:**

We have revised the sentences in the manuscript.

Page 6, Line 19, "3-methyltetrol sulfate ester is potentially chemically stable over its atmospheric lifetime."

**Comments #10**

*7,12-13 Note that this is only a likely mechanism.*

In the revised manuscript, we have reduced the mechanistic discussion and do not go into detailed reaction pathways such as the steric effects of functional groups on the formation of the reaction intermediates. We have removed the sentences in the revised manuscript.

**Comments #11**

*Figure 2 Y-axis should note m/z.*

**Author Response:**

Figure 2 illustrates the change in ion intensity detected as a function of OH exposure while *m/z* refers to the mass-to-charge-ratio of ions detected. We have modified the caption of Figure 2.

Figure 2, "The evolution of the ion intensity of bisulfate ion ($HSO_4^-$) as a function of OH exposure during the heterogeneous OH oxidation of potassium 3-methyltetrol sulfate ester at 70.8% RH."

**Comments #12**

*Figure 3 Y-axis should note m/z.*

**Author Response:**

Figure 3 illustrates the parent decay, which is the ion signal of 3-methyltetrol sulfate ester at a given OH exposure divided by the ion signal of 3-methyltetrol sulfate ester before oxidation, as a function of OH exposure. Hence, there would be no unit for the parent decay.

**Comments #13**

*Scheme 5. Only 1 pathway is shown, text seems to mention more than one.*

**Author Response:**

Scheme S5 shows the proposed reaction pathway of methyl sulfate anion decomposing into sulfate radical anion, as an extension of both Schemes S1 and S2 (as noted in Schemes S1 and S2).

---

## Author Comment (AC2) · 21 Jan 2019

*In this paper the authors describe results of a laboratory study of the heterogeneous oxidation of 3-methyltetrol sulfate ester by OH radicals in aqueous aerosol particles. This compound served as a surrogate for organosulfate esters formed from particle-phase reactions of isoprene oxidation products. Understanding the atmospheric fate of these compounds is important because of their potential importance as aerosol components. The reactions were conducted in a flow tube and the concentrations of the parent compound and products were monitored using DART mass spectrometry. The rates of oxidation were quantified using relative rate methods and the measured rate constant used to estimate the lifetime of these and similar compounds in the atmosphere due to this reaction. The only major product observed was bisulfate, and a mechanism was proposed to explain its presence. The paper is clearly and concisely written and the experiments are straightforward. The interpretation is reasonable, although it is challenging to build a believable mechanism when only one of the proposed products was detected. I think the paper should be publishable after some comments are addressed, but there is clearly work to be done to provide more convincing evidence for the proposed mechanism.*

**We would like to sincerely thank the reviewer for his/her thoughtful comments and suggestions. Please see our responses to reviewer's comments and suggestions below.**

**Comment #1**

*Page 2, line 34: The kinetics method assumes that the particles are uniformly mixed, which is probably true here. But in the atmosphere where particles contain numerous compounds with a range of surface-active properties there will be significant inhomogeneities in surface concentrations where the OH reaction occurs, thus affecting estimated lifetimes. Some comments on this are warranted.*

**Author Response:**

We agree with the reviewer that the chemical lifetime could be affected by surface-active compounds in the particle. If surface-active compounds are present, the chemical lifetime will likely be longer due to a lower surface concentration of parent molecules. This reduces the collision probability between gas-phase OH radicals and parent molecules at particle surface, leading to a smaller overall oxidation rate. We have added the following information in the revised manuscript.

Page 6, Line 20, "In the atmosphere where particles may contain compounds with different surface-active properties, the chemical lifetime could be affected due to the inhomogeneity in surface concentration. If surface-active compounds are present, the chemical lifetime will likely be longer due to a lower surface concentration of the parent molecules. This reduces the collision probability between gas-phase OH radicals and parent molecules at the particle surface, leading to a smaller overall oxidation rate."

**Comment #2**

*Page 5, line 20: The authors should justify their decision to ignore decomposition of α-hydroxyperoxy radicals to carbonyl + HO₂, which is the dominant pathway in the gas phase. This should be based on kinetics arguments since the absence of product peaks in the mass spectrum could be influenced by ionization. Relevant rate constants are available [Neta et al., J. Phys. Chem. Ref. Data, 19, 413-513 (1990)].*

**Author Response:**

Thanks for the comment. As suggested by the reviewer, the decomposition of α-hydroxylperoxy radical to carbonyl and $HO_2$ is a dominant pathway in gas phase. In our previous study, we have also found that this intramolecular $HO_2$ elimination process is an important reaction pathway during the heterogeneous OH oxidation of tartaric acid (Cheng et al., 2016). We do not include this reaction in this study because the carbonyl functionalization products have not been detected. Since the similarity in structures as compared to the parent compound (i.e. methyltetrol sulfate), we thought that the ionization issue might not be the primary reason for the absence of carbonyl products peak. However, as pointed by the reviewer, we agree that the detection of reaction products originated by the intramolecular $HO_2$ elimination process could be influenced by the ionization since the ionization efficiency and detection limit of the potential products are not fully understood. Further investigations on the ionization efficiency and detection limit of potential reaction products are needed to better understand the reaction pathways and the significance of the decomposition of α-hydroxyperoxy radicals through the $HO_2$ elimination process. In the revised manuscript, we have discussed and included this reaction pathway in the proposed reaction mechanisms in the supplementary material.

Supplementary material, Page 3, Line 27, "It is noted that decomposition of α-hydroxylperoxy radical to carbonyl and $HO_2$ is a dominant pathway in gas phase (Neta et al., 1990). In our previous study, we have also found that this intramolecular $HO_2$ elimination process is an important reaction pathway during the heterogeneous OH oxidation of tartaric acid (Cheng et al., 2016). However, the carbonyl functionalization products have not detected in the aerosol mass spectra. Given the similarity in structures as compared to the parent compound (i.e. methyltetrol sulfate), the ionization issue might not be the primary reason for the absence of carbonyl products in the mass spectra. However, further investigations on the ionization efficiency and detection limit of reaction products are needed to better understand the reaction pathways and the significance of the decomposition of α-hydroxyperoxy radicals through $HO_2$ elimination process during oxidation."

*References*

Cheng, C.T., Chan, M. N., Wilson, K.R.: Importance of unimolecular HO2 elimination in the heterogeneous OH reaction of highly oxygenated tartaric acid aerosol", J. Phy. Chem. A, 118, 28978–28992, 2016.

Neta, P., Huie, R. E., and Ross, A. B.: Rate constants for reactions of peroxyl radicals in fluid solutions, J. Phys. Chem. Ref. Data, 19, 413–513, 1990.

**Comment #3**

*Page 6, lines 13–18: I am not yet convinced by the argument that because the authors were able to detect small organosulfates in a previous study that they would have seen them here if they were present. How sensitive is the DART method to matrix effects, concentrations, and other conditions?*

**Author Response:**

We agree with the reviewer's concern. We acknowledge that the detection limits and possible matrix effects are not yet well understood. Further studies on the sensitivity of DART-MS to

smaller organosulfates under different conditions are warranted. We have added this information in the revised manuscript.

Supplementary material, Page 2, Line 36, "In our previous study, we have demonstrated that small organosulfates, such as sodium methyl sulfate ($CH_3SO_4Na$) and sodium ethyl sulfate ($C_2H_5SO_4Na$), the alcohol and carbonyl functionalization products of sodium ethyl sulfate can be detected by the DART ionization source (Kwong et al., 2018). If these small organosulfates are formed in a significant amount during the oxidation of methyltetrol sulfate, the DART ionization technique will likely be able to detect them. We acknowledge that the detection limits and possible matrix effects are not yet well understood. Further studies on the sensitivity of DART-MS to smaller organosulfates under different conditions are warranted."

**Comment #4**

*Page 7, lines 1–2 and elsewhere: Why wouldn't compounds that evaporate also be detected by DART, since the method for analyzing the particles involves first evaporating them at high temperature?*

**Author Response:**

As mentioned in the experimental section, the volatile fragmentation products or gas-phase species were removed by an activated charcoal denuder before entering the heater as well as ionization region, leaving only the particle-phase products (see Page 3, Line 22). We have clarified this point in the revised manuscript.

Page 3, Line 22, "An annular Carulite catalyst denuder and an activated charcoal denuder were used to remove ozone and gas-phase species from the aerosol stream leaving the reactor, respectively. As a result, only particle-phase products are detected. A fraction of the aerosol stream was sampled by a scanning mobility particle sizer (SMPS, TSI, CPC Model 3775, Classifier Model 3081) to measure the aerosol size and number distribution. The surface-weighted diameter of the aerosols was measured to be 225.9 ± 1.4 nm before oxidation. The remaining flow was then directed into a stainless steel tube heater at 380–400°C, where the temperature and aerosol residence time in the heater were sufficient to completely vaporize the aerosols."

**Technical Comments**

*Page 3, line 15: Should be "chromatograph".*

**Author Response:**

We have revised the sentence.

Page 3, Line 15, "By measuring the decay of hexane using a gas chromatograph coupled with a flame ionization detector, the OH exposure, which is an integral of gas-phase OH radical concentration and reaction time, can be calculated (Smith et al., 2009):"

---

## Author Response (AR2)

**Reply to editor's comment on "Heterogeneous OH Oxidation of Isoprene Epoxydiol-Derived Organosulfates: Kinetics, Chemistry and Formation of Inorganic Sulfate" by Hoi Ki Lam et al.**

**Comment #1**

*Page 1, line 18: I think that the methyltetrol sulfates are the most abundant in the boundary layer, but for the overall atmosphere it has been suggested that it may be glycolic acid sulfate (Liao et al. JGR 2015).*

**Author Response:**

Thanks for the comment. We have revised the sentence in the abstract and added the potential of future investigations on the transformation of glycolic acid sulfate in the conclusion.

Page 1, line 17: "Acid-catalyzed multiphase chemistry of epoxydiols formed from isoprene oxidation yields the most abundant organosulfates (i.e., methyltetrol sulfates) detected in atmospheric fine aerosols in the boundary layer."

Page 7, line 12: "Future investigations on the transformation of other organosulfates, for instance glycolic acid sulfate, which is the most abundant organosulfates for the overall atmosphere, are also desirable (Liao et al., 2015)."

Reference

Liao, J., Froyd, K. D., Murphy, D. M., Keutsch, F. N., Yu, G., Wennberg, P. O., St. Clair, J. M., Crounse, J. D., Wisthaler, A., Mikoviny, T., Jimenez, J. L., Campuzano-Jost, Pedro, Day, D. A., Hu, W., Ryerson, T. B., Pollack, I. B., Peischl, J., Anderson, B. E., Ziemba, L. D., Blake, D. R., Meinardi, S., and Diskin, G.: Airborne measurements of organosulfates over the continental U. S., J. Geophys. Res.-Atmos., 120, 2990–3005, 2015.

**Comment #2**

*Page 1, line 23: In line with some of the reviewer comments. The way it is written the readers will at this stage assume that reaction products were observed. I suggest eliminating the reaction products as argument in the abstract. I do not think this detracts from the main finding, which is the rate constant.*

**Author Response:**

We agree with the comment and eliminating the reaction products in the abstract.

Page 1, line 20: "As a result, we investigate the heterogeneous oxidation of aerosols consisting of potassium 3-methyltetrol sulfate ester ($C_5H_{11}SO_7K$) by gas-phase hydroxyl (OH) radicals at a relative humidity (RH) of 70.8 %."

**Comment #3**

*Page 1, line 27: Also, following up on one of the reviewer comments. The effective rate OH constant is given as it is usually for a gas-phase reaction. It may be useful to explain this in the abstract or provide a context as why it is given as gas-phase when it is a heterogeneous reaction.*

**Author Response:**

Thanks for the comment. We agree the effective OH rate constant is for a gas-phase reaction. In the abstract, we have used the effective "heterogeneous" OH rate constant, which describes it is a heterogeneous reaction.

**Comment #4**
*Page 1, line 28: Please state the assumed OH concentration for this (1.5E6?).*

**Author Response:**

Thanks for the comment. We have added the OH concentration used for determining the chemical lifetime.

Page 1, line 25: "Kinetic measurements reveal that the effective heterogeneous OH rate constant is measured to be $4.74 \pm 0.2 \times 10^{-13}$ $cm^3$ $molecule^{-1}$ $s^{-1}$ with a chemical lifetime against OH oxidation of $16.2 \pm 0.3$ days assuming an OH radical concentration of $1.5 \times 10^6$ molecules $cm^{-3}$."

**Comment #5**

*Page 1, line 31: "absence of functionalization processes" is a little awkward. One could argue that changing an alcohol to an aldehyde or acid, even if it is just formaldehyde and formic acid, is a functionalization process as it changes the functional groups, so perhaps the phrasing could be improved. I think the important aspect is that the results strongly suggest fragmentation, as stated in the next sentence.*

**Author Response:**

We agree with the comments and have revised the sentence.

Page 1, line 29: "Aerosol mass spectra only show an increase in the intensity of bisulfate ion ($HSO_4^-$) after oxidation, suggesting the importance of fragmentation processes."

**Comment #6**

*Page 2, line 11: I may be mistaken, but my impression was that the C5-alkene triols are not direct SOA constituents but rather decomposition products from the analytical techniques.*

**Author Response:**

We agree that $C_5$-alkene triols could be decomposition products from analytical methods and have removed the it from the sentence.

Page 2, line 10: "This multiphase chemical pathway is a key for the substantial production of isoprene-derived SOA constituents (e.g. 2-methyltetrols, organosulfates, 3-ethyltetrahydrofuran-3,4-diols and oligomers) within atmospheric fine particulate matter ($PM_{2.5}$) (Carlton et al., 2009; Froyd et al., 2010; Surratt et al., 2010; Lin et al., 2012)."

---

## Author Response (AR3)

*In this paper the authors describe results of a laboratory study of the heterogeneous oxidation of 3-methyltetrol sulfate ester by OH radicals in aqueous aerosol particles. This compound served as a surrogate for organosulfate esters formed from particle-phase reactions of isoprene oxidation products. Understanding the atmospheric fate of these compounds is important because of their potential importance as aerosol components. The reactions were conducted in a flow tube and the concentrations of the parent compound and products were monitored using DART mass spectrometry. The rates of oxidation were quantified using relative rate methods and the measured rate constant used to estimate the lifetime of these and similar compounds in the atmosphere due to this reaction. The only major product observed was bisulfate, and a mechanism was proposed to explain its presence. The paper is clearly and concisely written and the experiments are straightforward. The interpretation is reasonable, although it is challenging to build a believable mechanism when only one of the proposed products was detected. I think the paper should be publishable after some comments are addressed, but there is clearly work to be done to provide more convincing evidence for the proposed mechanism.*

**We would like to sincerely thank the reviewer for his/her thoughtful comments and suggestions. Please see our responses to reviewer's comments and suggestions below.**

**Comment #1**

*Page 2, line 34: The kinetics method assumes that the particles are uniformly mixed, which is probably true here. But in the atmosphere where particles contain numerous compounds with a range of surface-active properties there will be significant inhomogeneities in surface concentrations where the OH reaction occurs, thus affecting estimated lifetimes. Some comments on this are warranted.*

**Author Response:**

We agree with the reviewer that the chemical lifetime could be affected by surface-active compounds in the particle. If surface-active compounds are present, the chemical lifetime will likely be longer due to a lower surface concentration of parent molecules. This reduces the collision probability between gas-phase OH radicals and parent molecules at particle surface, leading to a smaller overall oxidation rate. We have added the following information in the revised manuscript.

Page 6, Line 20, "In the atmosphere where particles may contain compounds with different surface-active properties, the chemical lifetime could be affected due to the inhomogeneity in surface concentration. If surface-active compounds are present, the chemical lifetime will likely be longer due to a lower surface concentration of the parent molecules. This reduces the collision probability between gas-phase OH radicals and parent molecules at the particle surface, leading to a smaller overall oxidation rate."

**Comment #2**

*Page 5, line 20: The authors should justify their decision to ignore decomposition of a-hydroxyperoxy radicals to carbonyl + HO₂, which is the dominant pathway in the gas phase. This should be based on kinetics arguments since the absence of product peaks in the mass spectrum could be influenced by ionization. Relevant rate constants are available [Neta et al., J. Phys. Chem. Ref. Data, 19, 413-513 (1990)].*

**Author Response:**

Thanks for the comment. As suggested by the reviewer, the decomposition of α-hydroxylperoxy radical to carbonyl and $HO_2$ is a dominant pathway in gas phase. In our previous study, we have also found that this intramolecular $HO_2$ elimination process is an important reaction pathway during the heterogeneous OH oxidation of tartaric acid (Cheng et al., 2016). We do not include this reaction in this study because the carbonyl functionalization products have not been detected. Since the similarity in structures as compared to the parent compound (i.e. methyltetrol sulfate), we thought that the ionization issue might not be the primary reason for the absence of carbonyl products peak. However, as pointed by the reviewer, we agree that the detection of reaction products originated by the intramolecular $HO_2$ elimination process could be influenced by the ionization since the ionization efficiency and detection limit of the potential products are not fully understood. Further investigations on the ionization efficiency and detection limit of potential reaction products are needed to better understand the reaction pathways and the significance of the decomposition of α-hydroxyperoxy radicals through the $HO_2$ elimination process. In the revised manuscript, we have discussed and included this reaction pathway in the proposed reaction mechanisms in the supplementary material.

Supplementary material, Page 3, Line 27, "It is noted that decomposition of α-hydroxylperoxy radical to carbonyl and $HO_2$ is a dominant pathway in gas phase (Neta et al., 1990). In our previous study, we have also found that this intramolecular $HO_2$ elimination process is an important reaction pathway during the heterogeneous OH oxidation of tartaric acid (Cheng et al., 2016). However, the carbonyl functionalization products have not detected in the aerosol mass spectra. Given the similarity in structures as compared to the parent compound (i.e. methyltetrol sulfate), the ionization issue might not be the primary reason for the absence of carbonyl products in the mass spectra. However, further investigations on the ionization efficiency and detection limit of reaction products are needed to better understand the reaction pathways and the significance of the decomposition of α-hydroxyperoxy radicals through $HO_2$ elimination process during oxidation."

*References*

Cheng, C.T., Chan, M. N., Wilson, K.R.: Importance of unimolecular HO2 elimination in the heterogeneous OH reaction of highly oxygenated tartaric acid aerosol", J. Phy. Chem. A, 118, 28978–28992, 2016.

Neta, P., Huie, R. E., and Ross, A. B.: Rate constants for reactions of peroxyl radicals in fluid solutions, J. Phys. Chem. Ref. Data, 19, 413–513, 1990.

**Comment #3**

*Page 6, lines 13–18: I am not yet convinced by the argument that because the authors were able to detect small organosulfates in a previous study that they would have seen them here if they were present. How sensitive is the DART method to matrix effects, concentrations, and other conditions?*

**Author Response:**

We agree with the reviewer's concern. We acknowledge that the detection limits and possible matrix effects are not yet well understood. Further studies on the sensitivity of DART-MS to

smaller organosulfates under different conditions are warranted. We have added this information in the revised manuscript.

Supplementary material, Page 2, Line 36, "In our previous study, we have demonstrated that small organosulfates, such as sodium methyl sulfate ($CH_3SO_4Na$) and sodium ethyl sulfate ($C_2H_5SO_4Na$), the alcohol and carbonyl functionalization products of sodium ethyl sulfate can be detected by the DART ionization source (Kwong et al., 2018). If these small organosulfates are formed in a significant amount during the oxidation of methyltetrol sulfate, the DART ionization technique will likely be able to detect them. We acknowledge that the detection limits and possible matrix effects are not yet well understood. Further studies on the sensitivity of DART-MS to smaller organosulfates under different conditions are warranted."

**Comment #4**

*Page 7, lines 1–2 and elsewhere: Why wouldn't compounds that evaporate also be detected by DART, since the method for analyzing the particles involves first evaporating them at high temperature?*

**Author Response:**

As mentioned in the experimental section, the volatile fragmentation products or gas-phase species were removed by an activated charcoal denuder before entering the heater as well as ionization region, leaving only the particle-phase products (see Page 3, Line 22). We have clarified this point in the revised manuscript.

Page 3, Line 22, "An annular Carulite catalyst denuder and an activated charcoal denuder were used to remove ozone and gas-phase species from the aerosol stream leaving the reactor, respectively. As a result, only particle-phase products are detected. A fraction of the aerosol stream was sampled by a scanning mobility particle sizer (SMPS, TSI, CPC Model 3775, Classifier Model 3081) to measure the aerosol size and number distribution. The surface-weighted diameter of the aerosols was measured to be 225.9 ± 1.4 nm before oxidation. The remaining flow was then directed into a stainless steel tube heater at 380–400°C, where the temperature and aerosol residence time in the heater were sufficient to completely vaporize the aerosols."

**Technical Comments**

*Page 3, line 15: Should be "chromatograph".*

**Author Response:**

We have revised the sentence.

Page 3, Line 15, "By measuring the decay of hexane using a gas chromatograph coupled with a flame ionization detector, the OH exposure, which is an integral of gas-phase OH radical concentration and reaction time, can be calculated (Smith et al., 2009):"

**Reply to Interactive comment on "Heterogeneous OH Oxidation of Isoprene Epoxydiol-Derived Organosulfates: Kinetics, Chemistry and Formation of Inorganic Sulfate" by Hoi Ki Lam et al.**

**Anonymous Referee #3**

*This manuscript describes the OH oxidation of small organosulfates derived from atmospheric oxidation of isoprene. Using advanced mass spectrometry of aerosol particles generated in an aerosol flow reactor, it is shown that loss of 3-methyltetrol sulfate ester is slow, with a lifetime for oxidation of more than 2 weeks under likely atmospheric conditions. This suggests that other loss processes (e.g. deposition) may be the main atmospheric loss process for similar organosulfate esters. Lack of detection of smaller oxidation products suggests formation of volatile products upon oxidation. These results are consistent with relatively high observed levels of organosulfates in atmospheric aerosol. This work is a solid contribution to the field, and I recommend this manuscript for publication pending minor revisions.*

**We would like to sincerely thank the reviewer for his/her thoughtful comments and suggestions. Please see our responses to reviewer's comments and suggestions below.**

**Comment #1**

*The experimental setup and data are fairly straightforward, generally giving clear, direct support of the key conclusion of a long atmospheric lifetime for small, oxidized organosulfates, but the detailed and long discussion of mechanistic details is not warranted. Without detection, in either gas or aerosol phase, of any reaction products other than bisulfate, the mechanistic discussion is largely speculative. Nearly all of this discussion should be removed. All reaction schemes presented should be moved to supplementary material and the mechanistic discussion must be reduced significantly. You cannot make such strong claims about a mechanism in which you more or less only observe loss of product with inability to detect products. Table 1 is the crucial result of this paper, not the speculation about the mechanism for oxidation.*

**Author Response:**

We agree with the reviewer's comment. In the revised manuscript, we have reduced the mechanistic discussion and moved the detailed discussion and reaction schemes to the supplementary material. We also address that the reaction pathways (Schemes S1-S5) are tentatively proposed based on the observed aerosol-phase products detected by the DART. To better understand the reaction pathways, more studies are needed to characterize and quantify the reaction products in both aerosol and gas phases with different analytical techniques in order to cross check the aerosol composition data obtained by DART and verify the proposed reaction pathways in this work.

Page 5, Line 20,

" **3.3 Proposed Reaction Mechanisms**

Based on the aerosol mass spectra and well-known aerosol-phase reactions previously reported in the literature (George and Abbatt, 2010; Kroll et al., 2015), we tentatively propose reaction mechanisms for the heterogeneous OH oxidation of 3-methyltetrol sulfate ester. The reaction schemes proposed can be found in the supplementary materials (Schemes S1-S5). Briefly, potassium methyltetrol sulfate ester likely dissociates and exists in its ionic form in the droplets.

In the first oxidation step, the OH radical abstracts a hydrogen atom to form an alkyl radical which quickly reacts with oxygen to form a peroxy radical. We propose that the formation of alkoxy radical may be favored over the Russell mechanism (Russell, 1957) and Bennett–Summers reactions (Bennett and Summers, 1974) as functionalization products were not detected. Alkoxy radicals, once formed, may tend to undergo fragmentation due to the presence of vicinal hydroxyl groups, which lower the activation energy required for the decomposition of the alkoxy radicals (Cheng et al., 2015; Wiegel et al., 2015; Jimenez et al., 2009; Peeters et al., 2004; Vereecken and Peeters, 2009).

*Formation of $HSO_4^-$*. Sulfate radical anion ($SO_4^{\bullet-}$) can be formed through the decomposition of the alkoxy radical and is a strong oxidant in aqueous phase (Neta et al., 1988; Clifton and Huie, 1989; Padmaja et al., 1993). $SO_4^{\bullet-}$ can abstract a hydrogen atom from a neighboring organic molecule (e.g. unreacted 3-methyltetrol sulfate ester) to form $HSO_4^-$ (R1) or react with particle-phase water to yield a $HSO_4^-$ and an OH radical (R2) (Tang et al., 1988) as illustrated below. It is noted that $SO_4^{\bullet-}$ or OH radical recycled from R2 can react with 3-methyltetrol sulfate ester, contributing to the secondary chain reactions.

$$C_5H_{11}O_7S^- + SO_4^{\bullet-} \rightarrow C_5H_{10}O_7S^{\bullet-} + HSO_4^- \qquad\qquad (R1)$$

$$SO_4^{\bullet-} + H_2O \rightleftharpoons OH^{\bullet} + HSO_4^- \qquad\qquad (R2)$$

Since 3-methyltetrol sulfate ester is unlikely to hydrolyze (Darer et al., 2011), the formation of the $HSO_4^-$ upon OH oxidation could be best explained by the formation and subsequent reactions of $SO_4^{\bullet-}$.

Based on the proposed reaction mechanisms, the decomposition of alkoxy radicals can lead to formation fragmentation products (without sulfate group) and smaller organosulfates. We acknowledge that the ionization efficiency and detection limit of the reaction products are not fully understood. The absence of the potential products might attribute to the DART ionization and detection issues. More work is needed to investigate the formation and abundance of the reaction products formed in the aerosol and gas phases upon oxidation in order to better understand the reaction pathways. It is also noted that the formation of second or higher generation products are possible due to the high OH concentrations used in this study. While the potential secondary or higher generation products have not detected possibly due to their low concentrations and/or ionization issues, for clarify, the formation and oxidation of second or higher generation products are not further discussed."

**Comment #2**

*You must comment on the potential for side-reactions, such as oxidation of second-generation products, due to the high OH levels used in this study.*

**Author Response:**

We agree with the reviewer that first generation products can be further oxidized to form second or higher generation products due to the high OH levels in this study. However, the potential secondary or higher generation products have not detected possibly due to their low concentration and/or ionization efficiency. For clarity, the formation and oxidation of second or higher generation products are addressed but will not be discussed in details.

Page 6, Line 9, "It is also noted that the formation of second or higher generation products are possible due to the high OH concentrations used in this study. While the potential secondary or higher generation products have not detected possibly due to their low concentrations and/or ionization issues, for clarity, the formation and oxidation of second or higher generation products are not further discussed."

**Comment #3**

*Supplementary materials must include a figure showing time dependent variables during an experiment including: aerosol size distributions, hexane concentration, and product signal. This may be nearly a step function, but it should be clearly shown that your system clearly responds to initiation of oxidation.*

**Author Response:**

We would like to note that oxidation experiments were carried out in a flow tube reactor. We have measured the composition and size of the aerosols leaving the reactor before and after OH oxidation with a fixed reaction time (aerosol residence time is about 1.5 minutes in this study. Hence, the time dependent data (e.g. the time evolved data of aerosol size distributions and product signal) are not available in our study.

**Comments #4**

*It should be noted that volatile products formed in the aerosol used here might remain in other aerosol if the Henry's law constant changes dramatically with composition, particularly if the volatile products are highly reactive with other types of aerosol species not present in your study (e.g. reactive nitrogen species). Comment on precedents available in published literature.*

**Author Response:**

We agree with the reviewer that based on the proposed reaction mechanisms, volatile fragmentation products likely contain polar functional groups. They might partition back to the aerosols, for instance aqueous droplets because of their high water solubilities or Henry's law constants. Additionally, these fragmentation products could be reactively uptaken by the aerosols which contain reactive nitrogen or oxygen species through reactions. We have added this information in the revised manuscript.

Page 6, Line 29, "It is also noted that volatile fragmentation products likely contain polar functional groups. They may partition back to the aerosols, for instance aqueous droplets because of their high water solubilities or Henry's law constants. Additionally, they could be reactively uptaken by aerosols which contain reactive nitrogen or oxygen species through reactions."

**Comments #5**

*Due to the open ionization region, is it possible that you lose volatile products as a result of detection method? This would mean that non-detection of products might be caused by the method of detection rather than evaporation under reasonable atmospheric conditions.*

*The non-detection of products (particularly from functionalization) using DART would be good information to add to the abstract.*

**Author Response:**

As mentioned in the experimental section, the volatile fragmentation products or gas-phase species were removed by an activated charcoal denuder before entering the heater as well as ionization region, leaving only the particle-phase products (see Page 2 Line 22). We have clarified this point.

Page 3, Line 22, "An annular Carulite catalyst denuder and an activated charcoal denuder were used to remove ozone and gas-phase species from the aerosol stream leaving the reactor,

respectively. As a result, only particle-phase products are detected. A fraction of the aerosol stream was sampled by a scanning mobility particle sizer (SMPS, TSI, CPC Model 3775, Classifier Model 3081) to measure the aerosol size and number distribution. The surface-weighted diameter of the aerosols was measured to be $225.9 \pm 1.4$ nm before oxidation. The remaining flow was then directed into a stainless steel tube heater at 380–400°C, where the temperature and aerosol residence time in the heater were sufficient to completely vaporize the aerosols."

**Comments #6**

*Section 3.3 This should be reduced to a single paragraph that only mentions the proposed mechanism and does not go into details concerning issues such as the sterics of the reaction intermediates.*

**Author Response:**

The reviewer has raised the same concern in **Comment #1.** We agree with the reviewer's comment. In the revised manuscript, we have reduced mechanistic discussion and moved the reaction schemes to the supplementary materials. We also address that the reaction pathways (Schemes S1-S5) are tentatively proposed based on the observed products detected by the DART. Please see our response to **Comment #1**. We have also removed the content about the steric effects on the formation of reaction products from the Abstract.

**Comments #7**

*Page 2, Line 2 Change to read ", the largest atmospheric source"*

**Author Response:**

We have revised the sentence in the manuscript.

Page 2, Line 2, "Isoprene (2-methyl-1,3-butadiene, $C_5H_8$), emitted from terrestrial vegetation to the atmosphere, is the largest atmospheric source of non-methane volatile organic compounds."

**Comments #8**

*Page 4, Line 6, Change to read "The hygroscpocity"*

**Author Response:**

We have revised the sentence in the manuscript.

Page 4, Line 5, "The hygroscopicity of potassium 3-methyltetrol sulfate ester aerosols has not been experimentally determined."

**Comments #9**

*Page 7, Line 9, Change to read "stable over its atmospheric lifetime."*

**Author Response:**

We have revised the sentences in the manuscript.

Page 6, Line 19, "3-methyltetrol sulfate ester is potentially chemically stable over its atmospheric lifetime."

**Comments #10**

*7,12-13 Note that this is only a likely mechanism.*

In the revised manuscript, we have reduced the mechanistic discussion and do not go into detailed reaction pathways such as the steric effects of functional groups on the formation of the reaction intermediates. We have removed the sentences in the revised manuscript.

**Comments #11**

*Figure 2 Y-axis should note m/z.*

**Author Response:**

Figure 2 illustrates the change in ion intensity detected as a function of OH exposure while *m/z* refers to the mass-to-charge-ratio of ions detected. We have modified the caption of Figure 2.

Figure 2, "The evolution of the ion intensity of bisulfate ion ($HSO_4^-$) as a function of OH exposure during the heterogeneous OH oxidation of potassium 3-methyltetrol sulfate ester at 70.8% RH."

**Comments #12**

*Figure 3 Y-axis should note m/z.*

**Author Response:**

Figure 3 illustrates the parent decay, which is the ion signal of 3-methyltetrol sulfate ester at a given OH exposure divided by the ion signal of 3-methyltetrol sulfate ester before oxidation, as a function of OH exposure. Hence, there would be no unit for the parent decay.

**Comments #13**

*Scheme 5. Only 1 pathway is shown, text seems to mention more than one.*

**Author Response:**

Scheme S5 shows the proposed reaction pathway of methyl sulfate anion decomposing into sulfate radical anion, as an extension of both Schemes S1 and S2 (as noted in Schemes S1 and S2).

**Reply to comment on "Heterogeneous OH Oxidation of Isoprene Epoxydiol-Derived Organosulfates: Kinetics, Chemistry and Formation of Inorganic Sulfate" by Hoi Ki Lam et al.**

**Editor**

**Comment #1**

*Page 1, line 18: I think that the metyltetrol sulfates are the most abundant in the boundary layer, but for the overall atmosphere it has been suggested that it may be glycolic acid sulfate (Liao et al. JGR 2015).*

**Author Response:**

Thanks for the comment. We have revised the sentence and added the potential of future investigation on the transformation of glycolic acid sulfate in the conclusion.

Page 1, line 17: "Acid-catalyzed multiphase chemistry of epoxydiols formed from isoprene oxidation yields the most abundant organosulfates (i.e., methyltetrol sulfates) detected in fine aerosols in the boundary layer."

Page 7, line 12: "Future investigation on the transformation of other organosulfates, for instance glycolic acid sulfate, which is the most abundant organosulfates for the overall atmosphere, may also be desirable (Liao et al., 2015)."

**Comment #2**

*Page 1, line 23: In line with some of the reviewer comments. The way it is written the readers will at this stage assume that reaction products were observed. I suggest eliminating the reaction products as argument in the abstract. I do not think this detracts from the main finding, which is the rate constant.*

**Author Response:**

Thanks for the comment. We have revised the sentence.

Page 1, line 20: "As a result, we investigate the heterogeneous oxidation of aerosols consisting of potassium 3-methyltetrol sulfate ester ($C_5H_{11}SO_7K$) by gas-phase hydroxyl (OH) radicals at a relative humidity (RH) of 70.8 %."

**Comment #3**

*Page 1, line 27: Also, following up on one of the reviewer comments. The effective rate OH constant is given as it is usually for a gas-phase reaction. It may be useful to explain this in the abstract or provide a context as why it is given as gas-phase when it is a heterogeneous reaction.*

**Author Response:**

Thanks for the comment. The effective OH rate constant originates from considering the second order rate equation for the reaction between organic and 3-MGA:

$$\frac{d[3-MGA]}{dt} = -k[3-MGA][OH]$$

By integrating and assume the ionization efficiency of 3-MGA is 1, we have

$$ln\frac{I}{I_0} = -k \cdot [OH] \, t$$

The rate constant derived, as a result, should be applicable to heterogeneous reaction.

**Comment #4**

*Page 1, line 28: Please state the assumed OH concentration for this (1.5E6?).*

**Author Response:**

Thanks for the comment. We have added the assumed OH concentration.

Page 1, line 25: "Kinetic measurements reveal that the effective heterogeneous OH rate constant is measured to be $4.74 \pm 0.2 \times 10^{-13}$ $cm^3$ $molecule^{-1}$ $s^{-1}$ with a chemical lifetime against OH oxidation of $16.2 \pm 0.3$ days assuming an OH radical concentration of $1.5 \times 10^6$ molecules $cm^{-3}$."

**Comment #5**

*Page 1, line 31: "absence of functionalization processes" is a little awkward. One could argue that changing an alcohol to an aldehyde or acid, even if it is just formaldehyde and formic acid, is a functionalization process as it changes the functional groups, so perhaps the phrasing could be improved. I think the important aspect is that the results strongly suggest fragmentation, as stated in the next sentence.*

**Author Response:**

Thanks for the comment. We have revised the sentence.

Page 1, line 29: "Aerosol mass spectra only show an increase in the intensity of bisulfate ion ($HSO_4^-$) after oxidation, strongly suggesting fragmentation."

**Comment #6**

*Page 2, line 11: I may be mistaken, but my impression was that the C5-alkene triols are not direct SOA constituents but rather decomposition products from the analytical techniques.*

**Author Response:**

Thanks for the comment. We have removed the C5-alkene triols.

Page 2, line 9: "This multiphase chemical pathway is a key for the substantial production of isoprene-derived SOA constituents (e.g. 2-methyltetrols, C5-alkene triols, organosulfates, 3-

methyltetrahydrofuran-3,4-diols and oligomers) within atmospheric fine particulate matter (PM2.5) (Carlton et al., 2009; Froyd et al., 2010; Surratt et al., 2010; Lin et al., 2012)."